# The Inflammatory Lung Microenvironment; a Key Mediator in MSC Licensing

**DOI:** 10.3390/cells10112982

**Published:** 2021-11-02

**Authors:** Hazel Dunbar, Daniel J Weiss, Sara Rolandsson Enes, John G Laffey, Karen English

**Affiliations:** 1Department of Biology, Maynooth University, W23 F2H6 Maynooth, Ireland; hazel.dunbar.2015@mumail.ie; 2Kathleen Lonsdale Institute for Human Health Research, Maynooth University, W23 F2H6 Maynooth, Ireland; 3Department of Medicine, 226 Health Science Research Facility, Larner College of Medicine, University of Vermont, Burlington, VT 05405, USA; daniel.weiss@med.uvm.edu; 4Department of Experimental Medical Science, Faculty of Medicine, Lund University, 22100 Lund, Sweden; sara.rolandsson_enes@med.lu.se; 5Regenerative Medicine Institute (REMEDI) at CÚRAM Centre for Research in Medical Devices, Biomedical Sciences Building, National University of Ireland Galway, H91 W2TY Galway, Ireland; john.laffey@nuigalway.ie; 6Department of Anaesthesia, Galway University Hospitals, SAOLTA University Health Group, H91 YR71 Galway, Ireland

**Keywords:** MSC, licensing, lung, microenvironment, inflammatory, cytokines, immunomodulatory

## Abstract

Recent clinical trials of mesenchymal stromal cell (MSC) therapy for various inflammatory conditions have highlighted the significant benefit to patients who respond to MSC administration. Thus, there is strong interest in investigating MSC therapy in acute inflammatory lung conditions, such as acute respiratory distress syndrome (ARDS). Unfortunately, not all patients respond, and evidence now suggests that the differential disease microenvironment present across patients and sub-phenotypes of disease or across disease severities influences MSC licensing, function and therapeutic efficacy. Here, we discuss the importance of licensing MSCs and the need to better understand how the disease microenvironment influences MSC activation and therapeutic actions, in addition to the need for a patient-stratification approach.

## 1. Introduction

The morbidity and mortality associated with acute respiratory disease have never been more prominent than during the COVID-19 pandemic. In particular, the lack of therapeutics for treating lung inflammatory conditions including acute respiratory distress syndrome (ARDS) have highlighted the urgent unmet need for new therapeutic approaches. Mesenchymal stromal cells (MSCs) derived from both mouse and human bone marrow (BM), human umbilical cord (UC), adipose tissue (AT) and amniotic (A) tissue have shown positive outcomes in a broad spectrum of lung diseases [1] in preclinical studies, including asthma [2,3], idiopathic pulmonary fibrosis (IPF) [4,5,6], chronic obstructive pulmonary disease (COPD) [7,8], acute lung injury (ALI) [9,10,11], and acute respiratory distress syndrome (ARDS) [12,13,14]. Clinical trials have investigated MSC therapy (BM and UC) in IPF [15], COPD [16,17,18], ARDS [19,20] and in COVID-19 associated ARDS [21] showing safety but have not yet shown efficacy. While MSCs (BM or AT) have been approved in some countries for use in treating acute graft versus host disease (aGvHD) [22,23,24] and for Crohn’s fistula [25,26], MSC therapy has not yet been approved for lung inflammatory disorders [27]. However, MSCs have a proven safety profile in clinical trials [27] for inflammatory lung disorders and there are currently 151 clinical trials investigating MSCs as a lung intervention (https://ClinicalTrials.gov/, accessed on 5 October 2021), many of these for COVID-19-associated ARDS. While significant progress has been made in understanding the mechanisms by which MSC mediate their anti-inflammatory and pro-reparative effects in vitro, there are gaps in our understanding of how MSCs mediate their therapeutic effects in vivo [28,29]. A significant body of work has clearly identified the requirement for MSCs to be activated or licensed by signals such as pro-inflammatory cytokines to mediate MSC therapeutic effects [28,30]. It is becoming increasingly apparent that the disease inflammatory environment into which MSCs are administered is of critical importance for MSC capacity to suppress inflammation and promote repair [27,28]. The success of clinical application of MSC administration is limited by unclear understanding of the role of the microenvironment on MSC capacity to suppress inflammation and modulate immune responses. In the context of inflammatory lung disorders such as ARDS, there is now a general consensus that hyper- and hypo- inflammatory traits can be identified in ARDS patients [31]. Similarly, there are significant differences between the lung microenvironments of mild, moderate and severe asthmatics [32]. Thus, a better understanding of how the disease microenvironment may influence MSC therapeutic efficacy would help to identify the patients in which MSC therapy may be of most benefit. Herein, we provide an overview of our current understanding of how the disease microenvironment impacts MSC therapeutic efficacy and potential strategies to license or activate the anti-inflammatory and pro-reparative functions of MSCs. Where possible, we will focus on studies in inflammatory lung disease as well as evidence from other inflammatory conditions that might help us to better understand the role of the disease microenvironment and the potential for licensing to impact on MSC protective effects and therapeutic efficacy.

## 2. The Progress in the Clinical Translation of MSCs for Inflammatory Lung Disease

While earlier studies have not provided clear evidence of efficacy in phase 1/2 randomised controlled trials of MSCs in COPD [17] and ARDS [19,20] there are some reports of positive effects. A phase 2 trial enrolling 62 randomised patients with COPD, deemed the systemic administration of BM-MSC safe, although there were no differences in pulmonary function testing or with the 6-min walking test. However, a decrease in C-reactive protein was observed in comparison to elevated C-reactive protein (CRP) levels upon study entry (NCT00683722) [17]. In the context of the growing body of research suggesting the importance of the inflammatory lung profile of patients and its role in activating or licensing MSCs, a post-hoc analysis of the trial data was performed with stratification of COPD patients based on baseline levels of circulating inflammatory marker CRP. Interestingly, the data demonstrated that Remestemcel-L (BM-MSC) provided significant improvements in forced expiratory volume in one second, forced vital capacity and six-minute walk distance at 120 days post-infusion in patients with a higher baseline CRP [18]. In the context of ARDS, the findings from MSC clinical trials have not been clear-cut with respect to efficacy but studies have had favourable outcomes. Many of the current clinical trials investigating MSCs in lung inflammatory conditions are in COVID-19 ARDS and there is now a growing body of literature supportive of MSC efficacy in both non COVID-19 ARDS and in COVID-19 ARDS. A double-blinded randomised phase 2a safety trial investigating the use of allogeneic bone marrow derived-MSCs (BM-MSC) in severe ARDS patients showed there was no infusion-related haemodynamic or respiratory adverse events, proving their safety, although, the data could not support a claim for MSC efficacy in this trial and that may have been associated with reduced viability of the cell therapy product (NCT02097641). However, patients infused with BM-MSCs (with higher viability post-thaw) had lower concentrations of angiopoietin 2 (Ang-2) in their plasma after 6 h [19]. Interestingly, a nested cohort study within a phase 2a trial investigating BM-MSCs for moderate-to-severe ARDS, demonstrated that MSC treatment significantly reduced airspace total protein, Ang-2, IL-6 and soluble tumour necrosis factor (TNF) receptor-1 concentrations within a 48 h window following administration (NCT02097641) [33]. In addition, a phase 1 study of UC-MSCs in moderate-to-severe ARDS showed safety and reduction of circulating inflammatory biomarkers (ISRCTN52319075) [34].

Positive findings from the completed Athersys MUST-ARDS phase 1/2 randomised, double blind, placebo-controlled exploratory clinical study of MultiStem^®^ (BM derived human MSC-like cells) therapy in ARDS have been reported in a published conference abstract [35]. Interestingly, MultiStem® therapy enhanced ventilator-free days and ICU-free days and reduced mortality [35]. Ricordi and colleagues have also published the findings from their double-blind randomised control phase 1/2a trial of UC-MSCs in COVID-19 ARDS reporting significantly improved patient survival and significant decreases in pro-inflammatory cytokines in UC-MSC treated subjects at day 6 (NCT04355728) [21]. Similarly, a randomised controlled clinical trial investigating UC-MSCs reported improved survival rate, reduced length of stay and ventilator use as well as a decrease in IL-6 in patients who received UC-MSCs in COVID-19 ARDS (NCT04457609) [36]. Moreover, Mesoblast have reported positive initial findings from their phase 3 randomised, double-blind, placebo-controlled trial investigating Remestemcel-L in COVID-19 ARDS at international conferences (NCT04371393) [37]. Importantly, the work from Calfee and colleagues and others supports the idea of identifying phenotypes of ARDS [38,39] or treatable traits [40] and using that information to facilitate a personalised medicine approach. Together these studies suggest that patient stratification to identify disease phenotypes that might best respond to MSC therapy may increase the chance for MSC therapeutic efficacy.

## 3. The Importance of the Inflammatory Disease Microenvironment on MSC Therapeutic Efficacy

Pre-clinical studies have shown that the microenvironment present at the time of MSC administration influences MSCs’ actions and thus potential therapeutic efficacy (Figure 1). For example, if MSCs are administered too early before disease onset (such as in GvHD) [41,42,43], or if signals required for their immunomodulation (such as NFκB, TNF-α receptor, or IFN-γ) [41,44] are blocked, MSCs lose their protective effects. Activation or licensing of MSC immunomodulation often involves activation of NF-κB and NF-κB regulated genes. In situations, where pro-inflammatory cytokines are limited or NF-κB is inhibited, MSC efficacy is impaired [42,43,45]. Moreover, in microenvironments which prevent MSC immunomodulation or promote MSC death, the presence of allogeneic MSCs may even promote harm [29,44,46]. For example, MSCs promoted fibrotic changes and inhibited re-epithelialization in an acid-induced ALI model with high levels of oxidative stress, whereby the conditions present in this model either negatively impacted the MSCs or prevented their ability to immunomodulate and promote repair [47].

Notably, expression of PPARδ in mouse MSCs has been shown to impair MSC efficacy in a mouse model of arthritis, while knockdown or antagonism of PPARδ-enhanced mouse MSC efficacy via increased nitric oxide (NO) production [48]. In a humanised mouse model of acute GvHD, agonism of PPARδ in human BM-derived MSCs significantly impaired MSC therapeutic efficacy [49]. With respect to pre-clinical lung injury and sepsis models, upregulation of PPARδ signalling has been shown to play an important role in LPS-induced ALI and in caecal ligation puncture-induced sepsis [50,51]. Notably, angiopoietin-like protein 4 (Angptl4), a known target gene of PPARδ is upregulated in LPS-induced ALI [52] and has recently been reported as a clinical biomarker for ARDS [53]. Together, this evidence suggests that PPARδ ligands (agonists) are present in LPS-induced ALI which may contribute to microenvironmental impact on MSC therapeutic efficacy.

## 4. Licensing of MSCs Enhances Their Therapeutic Efficacy

IFN-γ, TNF-α and IL-1β have been identified as key mediators that facilitate the activation of MSC immunomodulation, in vitro [54,55,56,57]. A number of pre-clinical studies have demonstrated the capacity to enhance MSC therapeutic effects using a licensing approach in lung diseases including ALI, asthma and IPF (Table 1, Figure 1). Moreover, there is much we can learn about the mechanistic effects of licensed MSCs in other disease models, particularly where licensed MSCs have been administered systemically via the tail vein (Table 2). The licensing of mouse MSCs [41] or human BM-MSCs with IFN-γ before administration, enhanced MSC therapeutic efficacy in a mouse model [41] and a humanised mouse model of GvHD [43,49,54]. IFN-γ licensing of human BM-MSCs also enhanced therapeutic efficacy in pre-clinical models of TNBS-induced colitis and DSS-induced colitis [58]. Importantly, MSCs require stimulation with pro-inflammatory cytokines to produce immunomodulatory secreted factors like prostaglandin-E2 (PGE-2) [55,59], indolamine-2,3-dioxygenase (IDO) [60] and TNFα-stimulated gene 6 (TSG-6) [61], responsible for MSC therapeutic efficacy. TNF-α can act as an adjuvant for IFN-γ, where the two pro-inflammatory cytokines can work synergistically [62]. Licensing MSCs with TNF-α alone, enhances the production or secretion of factors such as PGE2, IDO, HGF and TSG-6 [55,63,64,65] and has been shown to enhance MSC efficacy in a number of disease models (reviewed in [66]).

Trophic factors, including those contained within extracellular vesicles (EVs) produced by MSCs, are thought to play a key role in mediating MSC therapeutics effects. EVs can transfer therapeutic cargo, such as mRNA, miRNA and even organelles like mitochondria, to ameliorate lung injury [70,80,81,82]. EVs are of particular interest, as they have been identified as a main effector of MSC paracrine function, playing a pivotal role for intracellular communication and boasting a therapeutic effect equivalent to that of their parent cells, MSCs [81]. Along with boosting MSC therapeutic potential, licensing with inflammatory cytokines can also boost EV efficacy [70,83,84]. The use of human and mouse BM- and human UC-MSC-derived EVs has been documented in pre-clinical models of lung disease, such as neonatal chronic lung disease [85], pneumonia [70,86,87], allergic asthma [3,88] and ALI [74,89]. EVs secreted from UC-MSCs and licensed with IFN-γ were more effective in attenuating *E. coli*-induced injury compared with EV from unlicensed UC-MSC in a rat model of *E. coli* pneumonia [70]. Furthermore, IFN-γ licensed UC-MSC EVs, but not naïve UC-MSC EVs, had the ability to reduce lung protein permeability, alveolar inflammation and alveolar-arterial oxygen gradient in injured lungs compared with controls [70].

Other cytokine licensing approaches include the use of oncostatin M and TGF-β. Licensing of BM-MSCs with oncostatin M enhanced MSC therapeutic efficacy in a bleomycin-induced fibrosis model [90], while TGF-β licensing of MSC enhanced MSC survival in a rat ALI model [69]. TGF-β-licensed mouse BM-MSCs have also been utilised in a corneal allograft mouse model, where they modulated the immune response by suppressing the effector T cell population and induced Tregs within the lung following intravenous administration [77].

‘Multi-cytokine licensing’, or ‘composite priming’ involving a cocktail of cytokines, including combinations of IFN-γ, TNF-α and IL-1β for MSC licensing, has also been investigated [43,55]. In this context, TNF-α and IL-1β increased the MSC expression of the IFN-γ receptor, enhancing the MSC immunoregulatory effects; thus, IL-1β can optimise the therapeutic effects initiated by MSC licensing with IFN-γ/TNF-α [91]. In line with this, monocyte-derived IL-1β activation of multipotent adult progenitor cells (MAPCs) was required for MAPC suppression of IL-7-induced CD4 T-cell proliferation [92]. In vivo, rat BM-MSCs, pre-licensed with TNF-α and IL-1β, promoted corneal allograft survival via myeloid cell-mediated induction of regulatory T cells in the lung [78].

In addition to cytokines, pre-conditioning, or licensing MSCs with toll-like receptor (TLR) ligands can enhance therapeutic efficacy in a wide array of inflammatory diseases. MSCs express a number of TLRs [93]. MSCs licensed with the TLR3 ligand Poly(I:C) provided enhanced therapeutic effects in pre-clinical models of TNBS-induced colitis [94], in cardiomyopathic hamsters [95] and in atopic dermatitis [96]. Licensed MSC-derived EVs via the TLR3 ligand Poly I:C, also exhibited beneficial effects with enhanced antimicrobial activity in pre-clinical mouse and ex-vivo-perfused human lung injured with severe E. coli pneumonia [87,97]. TLR4 priming of MSCs via LPS enhanced MSC efficacy in an experimental autoimmune encephalitis (EAE) model [98], while LPS-primed MSC-derived exosomes provided accelerated wound healing in a diabetic cutaneous wound model [99]. Using an alternative approach, Yu and Chiang utilised the TLR2 agonist Pam3CSK4 to license mouse BM-MSCs and showed that TLR2 activated mouse MSCs further decreased eosinophil infiltration in the lung and IL-4/IL-5 secretion in the bronchoalveolar lavage fluid (BALF) and had a greater impact on lung function compared to MSCs in an OVA-induced allergic airway model [100]. Early work from Waterman et al. highlighted the role of differential TLR ligation in driving pro- or anti-inflammatory activation of human MSCs [93]. Pre-conditioning with differential TLR agonists can modulate the MSC secretome, where TLR4 activation with LPS enhances secretion of pro-inflammatory mediators and TLR3 activation with Poly(I:C) increases secretion of immunosuppressive factors such as IDO and PGE2 [101,102]. Ligation of TLR2, but not TLR4, inhibited the chemotaxis of murine BM-MSC and reduced their ability to expand Treg populations in vitro [103]. Moreover, murine BM-MSCs licensed with TLR4 and IFN-γ alleviated liver fibrosis in mice infected with S. japonicum cercariae, compared with BM-MSCs licensed with TLR2 and IFN-γ, which exacerbated the immunopathology in vivo [104]. In a mouse model of experimental autoimmune encephalomyelitis (EAE), BM-MSCs licensed with Poly(I:C) reduced the proliferation of CD3^+^ T cells, compared with BM-MSCs licensed with LPS, which increased CD3^+^ T-cell proliferation. Following i.p. administration, Poly(I:C)-licensed BM-MSCs alleviated EAE severity in contrast to LPS-licensed BM-MSCs, where their immunosuppressive effects were reversed [105]. The differential effects of various TLR ligands on MSCs may be associated with the downstream activation of pro- or anti-inflammatory mediators by MSCs. It is also likely that differential disease microenvironments will alter TLR activation of MSCs in vivo.

The oxygen concentration in ex-vivo culture can have a significant impact on MSC function, particularly when there is a large difference between the oxygen concentration used during ex-vivo culture (usually normoxic) and the concentration available at the site of in vivo administration in inflammatory disease (usually hypoxic). Pre-conditioning in a hypoxic environment can enhance MSC survival [106,107,108,109,110] and MSCs’ secretion of trophic factors associated with their therapeutic efficacy [110,111,112,113]. Hypoxic licensing is of relevance, as a disruption in oxygen homeostasis results in a hypoxic environment in many inflammatory lung diseases [114], such as ARDS, wherein gas exchange is impaired [115,116]. Hypoxic pre-conditioning of MSCs enhanced MSC survival and therapeutic efficacy in bleomycin-induced lung fibrosis [67]. This enhanced survival was found to be partially linked to an upregulation in hepatocyte growth factor (HGF) [4]. The importance of HGF’s role in MSC cytoprotection has also been demonstrated using shRNA knockdown of HGF in human MSCs in a bleomycin-induced IPF model [4] and in an elastase-induced COPD model [8]. Differentially, MSCs exposed to hyperoxia (95% oxygen) can have an enhanced paracrine effect when administered to rats with oxygen-induced neonatal lung injury, due to an increase in stannocalcin-1 (STC-1) [117]. STC-1 has been described as having an important role in anti-apoptotic effects when secreted by MSCs [118]. UC-MSC derived microvesicles had enhanced angiogenesis potential following licensing with hypoxia, both in vitro and in vivo [119]. Alternatively, MSCs can also be pre-conditioned by culturing in anoxia, where ischemic MSC derived-exosomes have been shown to have enhanced protection in endotoxin-induced ALI in mice [120].

## 5. Exogenous Licensing of MSCs

There is growing evidence that different lung inflammatory environments, illustrated by utilising serum and BALF collected from various different inflammatory lung conditions as a surrogate, leads to altered MSC behaviours [2,47,121,122]. For example, ex-vivo exposure of murine MSCs to BALF or serum from mice with house dust mite (HDM)-induced allergic airway inflammation promoted increased expression of anti-inflammatory mediators (IDO, IL1RN, TSG-6, IL-10, TGF-β) and enhanced MSC therapeutic efficacy in HDM-mediated allergic airway inflammation [2]. Pre-conditioning of human MSCs with ARDS patient (moderately severe ARDS secondary to bacterial pneumonia) serum led to enhanced production of IL-10 and IL-1RN and decreased production of IL-6, IL-1 and IL-8 [73]. In contrast, BALF from patients with cystic fibrosis had toxic effects on human BM-MSCs, mediated by Aspergillus species-induced mitochondrial dysfunction and MSC death [123]. In the context of BALF from ARDS patients, a range of pro and anti-inflammatory mediators are induced in human MSCs following exposure to ARDS BALF (taken from ARDS patients without sepsis). Of particular interest, IL-1β present in these ARDS BALF samples was predictive of the induction of IL-6, IL-8 and FAS by human BM-MSCs following ex-vivo exposure to the BALF [124]. MSCs exposed to ARDS patient BALF samples were less effective at driving an anti-inflammatory macrophage phenotype compared to MSCs exposed to BALF from other lung conditions (including acute exacerbations of CF) [121]. A different study used pooled ARDS BALF and following exposure to these samples, human BM-MSCs promoted an anti-inflammatory and phagocytic macrophage phenotype, in vitro [74]. The limitations of these studies are that the ARDS patient etiologic profiles were not well described, and the use of different experimental conditions make it difficult to compare the findings from these studies. However, these studies highlight that the disease microenvironment present in the lung has the potential to have a significant positive or indeed negative effect on MSC therapeutic effects. As such, this underscores the need to better understand the disease microenvironment, how it influences MSC efficacy, and the potential benefits associated with a patient stratification approach to identify the patients who are most likely to respond to MSC therapy.

## 6. Endogenous Licensing of MSCs

Thus far, we have discussed the evidence supporting the fact that MSCs can influence their microenvironment once administered in vivo and the positive impact that exogenous licensing can have on MSC therapeutic effects. The previous paragraph alluded to the fact that MSCs can also be influenced by their microenvironment, for example, ex vivo, following exposure to BALF or serum from inflammatory conditions. Based on the status of the microenvironment, MSCs can either have a beneficial or detrimental effect in the case of acute lung injury [47]. One of the earliest studies focusing on the microenvironmental effects on MSC efficacy showed that MSCs failed to modulate the immune response in GvHD driven by IFN-γ knock out T cells, demonstrating the importance of IFN-γ for activation of MSC immunomodulatory function in vivo [41]. In alignment with this, protective effects of BM-MSC administration in a mouse model of allergic airway inflammation were lost in IFN-γ receptor knockout mice [125]. While systemic administration of mouse BM-MSCs protected against ventilator-induced lung injury in mice, the same MSCs exacerbated injury in an acid-induced ALI mouse model [47]. Exacerbated injury was associated with higher BALF concentrations of IL-6, fibronectin, and lower levels of total antioxidant capacity (TAC) [47], highlighting conditions which negatively impact MSC functions. The proteomic profile of more than half of a cohort of ARDS (severe pneumonia without sepsis) patients serum samples (n = 33) shared this profile of high plasma fibronectin and low levels of TAC, suggesting that the microenvironment present in these patients may not be optimal for MSC administration [47]. This study from Islam et al. was an important study in highlighting that there may be lung inflammatory conditions that may not be compatible with MSC therapeutic efficacy. Notably, a recent clinical trial of BM-MSC in ARDS (inclusion criteria: sepsis with/out pneumonia, pneumonia without sepsis, aspiration only) showed a non-significant trend of higher 28-day mortality in patients after treatment with MSCs compared to that of placebo (NCT02097641) [19]. This may be associated with baseline imbalances in the severity of illness and low viability of the MSCs utilized in this trial, however, it is also possible that the microenvironment present in some of these patients was sub-optimal for MSC therapeutic efficacy. These investigators are now conducting another trial in a select population of patients with ARDS resulting from trauma as opposed to sepsis or pneumonia to better evaluate this issue. Importantly, there are also many pre-clinical studies that have identified ways in which the lung inflammatory environment facilitates the licensing or activation of MSCs. In addition to the positive effects associated with endogenous production or presence of NF-κB, TNF-α receptor, or IFN-γ [41,44,45], endogenous TNF-α has also been shown to play a key role in licensing MSCs, in vivo, through the induction of TSG-6. Human MSCs expressing high levels of TSG-6 improved survival and preserved body weight in a murine bleomycin model, compared with the control. Similarly, human MSCs attenuated LPS-induced inflammation in the lung via secretion of TSG-6, as knock down of TSG-6 expression abrogated the human BM-MSCs’ anti-inflammatory effects in this murine model of ALI [126]. Moreover, entrapment of BM-MSCs in the lung leads to their activation and production of TSG-6, which has been shown to play a role in protection against myocardial infarction in mice [64]. Endogenous TNF-α or TNF receptors present in the disease microenvironment have been shown to play a key role in MSC efficacy in pre-clinical models of sepsis [127] and cardiomyopathy [128]. BM-MSCs from TNF-α or TNF-R1 knockout mice did not protect against caecal ligation and puncture-induced sepsis following intravenous administration [79]. An elegant study identified an important role for induced pluripotent stem cells (iPSC)-MSCs sensitivity to endogenous TNF-α in protection mediated via mitochondrial transfer in anthracycline-induced cardiomyopathy [128]. Specifically, transplantation of iPSC-MSCs but not TNFαIP2 (TNF-α induced protein that regulates tunnelling nanotube (TNT) formation) knockdown iPSC-MSCs protected against cardiomyopathy [128].

In addition to pro-inflammatory cytokines, toll-like receptors (TLR) and their ligands also influence MSC efficacy and function in vivo. TLR4 knockout mouse BM-MSCs failed to protect against *E. coli*-induced pneumonia [10] and were not efficacious in pre-clinical models of EAE [98] in comparison with wildtype MSCs. On the other hand, the microenvironment in the failing heart or myocardial infarct promoted a pro-inflammatory phenotype in both resident and transplanted mouse MSCs via TLR4 activation [129]. Interestingly, TLR4 knockout MSCs maintained their expression of CD47 (a “don’t eat me” signal), increasing their survival and facilitating their protective effects in a pre-clinical model of myocardial infarction [129].

Some of the most interesting studies focused on understanding how MSCs mediate their effect, have identified an important role for host macrophages present at the site of administration. A growing body of recent literature suggests that MSC–macrophage crosstalk plays a key role [130,131] in shaping MSC anti-inflammatory effects. An increase in anti-inflammatory or non-classical monocytes or macrophages has been reported in a range of disease models following MSC administration [79,82,92,132,133,134,135,136]. With respect to mechanisms, the transfer of mitochondria from MSCs to macrophages has been shown to enhance their bioenergetics [84] and promote an anti-inflammatory phenotype [11,82]. Use of clodronate liposomes to delete macrophages in vivo has demonstrated that the presence of macrophages are essential for MSC-mediated anti-inflammatory effects in preclinical models of ARDS, corneal allo-transplantation, liver injury and DSS-induced colitis [11,133,137,138]. Phagocytosis of MSCs [139,140], MSC-EVs [141], MSC cytoplasm [142] or MSC mitochondrial transfer via tunnelling nanotubes [11] can drive anti-inflammatory macrophage phenotypes. Building complexity upon those findings, one study showed that macrophage phagocytosis of MSCs that have been killed by cytotoxic T cells plays a key role in human BM-MSC protection against GvHD following i.v. administration [76]. Importantly, i.v. administration of apoptotic MSCs did not have the same level of protection as live MSCs [76]. Moreover, while live MSCs were effective in combination with immunosuppressive drugs, heat-killed MSCs were not efficacious in pre-clinical allogeneic heart transplantation [143]. Many of these findings highlighting the importance of the MSC-macrophage crosstalk in driving MSC activation and therapeutic effects are in studies using systemic administration of MSCs, whereby MSCs become trapped in the lung and mediate their effects even in inflammatory conditions distal to the lung. Thus, if we are to better understand exactly how the disease microenvironment influences MSC licensing and therapeutic efficacy and to identify the mechanism used by MSCs in mediating their effects, then we need to include in-depth studies of the lung environments present before and after MSC administration when MSCs are administered intravenously.

## 7. Patient Stratification to Identify Responders to MSC-Based Therapy

MSCs are usually detected for only a short time (72 h) in the lung or any other organ following systemic administration [29,140,144], however, their longevity can be enhanced in an injured lung or in licensed MSCs. Interestingly, exposure to healthy control BALF promotes human BM-MSC expression of HLA-DR, arguably increasing recognition and clearance of the MSCs [124]. This doesn’t occur with ARDS BALF exposure, suggesting that the ARDS inflammatory environment may be protective of MSC survival [124]. We have also demonstrated that IFN-γ licensing of human BM-MSCs enhances their longevity in the short-term, in vivo, while pre-exposure to a PPARδ agonist significantly reduces MSC longevity in vivo in a humanised mouse model of aGvHD [49]. Despite the short time-frame, MSCs can mediate significant protective effects when administered to conditions where there is an acute inflammatory insult such as in ARDS. Data from chronic lung patients where MSCs have been investigated within clinical trials but have not demonstrated efficacy in COPD [17,145,146,147], or IPF [15,148]. Furthermore, preclinical evidence suggests that MSCs cannot promote the regeneration of fibrotic tissue when administered during established bleomycin-induced IPF [4,149,150] and delayed administration of MSC in an elastase-induced COPD model reduced MSC efficacy. Given that patients with IPF and COPD are likely to receive MSCs at a time when the disease is fully established, this data suggests that MSCs may not be efficacious. In the context of the acute nature of ARDS [151], and the growing body of literature supporting the potential for MSC efficacy in ARDS [19,21,35,36], it seems sensible that MSCs may be most suitable in acute inflammatory conditions. Moreover, the data discussed in this review also supports the idea that differential disease microenvironments present in some diseases or sub-phenotypes of disease may be better suited to facilitate MSC activation and lead to optimal MSC therapeutic effects. For example, stratification of ARDS patients based on the hyper and hypo-inflammatory phenotypes may lead to the identification of responders to MSC therapy. Similarly, COPD patient stratification based on CRP baseline levels may enhance MSC efficacy, suggesting that even in some chronic diseases, the inflammatory environment may dictate potential MSC efficacy. Moreover, the potential to enhance MSC efficacy via pre-conditioning or licensing before administration to patients may provide a solution to try to enhance MSC therapy in heterogeneous patients where MSCs may not receive activation signals, negatively influencing their efficacy. Interestingly, Horwitz and colleagues have registered a phase I clinical trial investigating IFN-γ-licensed MSCs as a prophylaxis against aGvHD (NCT04328714). The findings from this study are eagerly awaited as the first study evaluating licensed MSCs in clinical trials.

## 8. Conclusions

In patients who respond to MSC therapy, these cells can have significant effects on the morbidity and in some cases mortality of patients who are very unwell and who have limited options. Although it may seem that limited progress has been made in the translation of MSC therapy to patients with inflammatory conditions (particularly in the lung), the field has learned much about how these cells respond to the inflammatory/disease microenvironment in which they find themselves following administration. Considering this, there is also a significant volume of research to be done in order for us to fully appreciate and understand how best to utilise these cells, so that we can identify the mechanisms of action and critical quality attributes required by the regulatory agencies. In our opinion, significant efforts should be made at the pre-clinical model stage and in patients following MSC administration to better identify how the disease microenvironment influences MSC licensing, function and efficacy.

## Figures and Tables

**Figure 1 cells-10-02982-f001:**
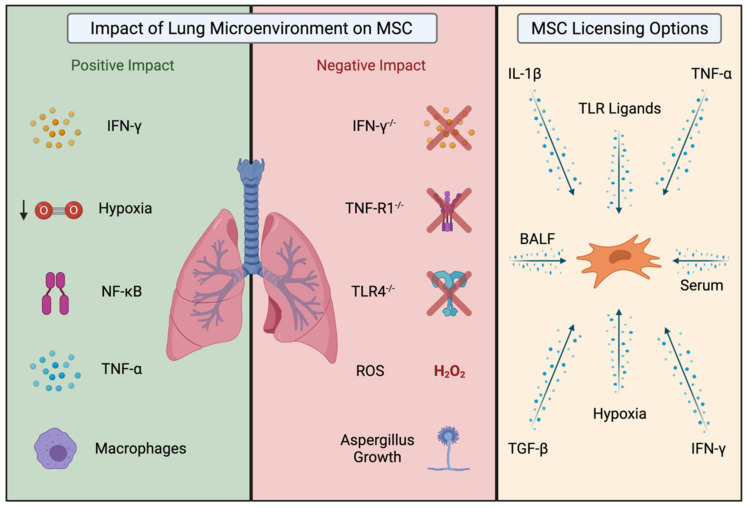
The Lung Microenvironment can positively or negatively influence MSC licensing and therapeutic effects. The presence of pro-inflammatory cytokines IFN-γ and TNF-α license MSCs and promote MSC immunomodulation. A hypoxic environment can license MSCs and enhance their survival. Activation of NFκB signalling has been identified as an important factor in MSC licensing. The crosstalk or interaction between MSCs and macrophages following intravenous administration has been shown to play a key role in shaping MSC therapeutic effects. The absence of TLR4, TNF-R1 or IFN-γ on MSC or in the disease microenvironment has been shown to negatively impact MSC function. The presence of Aspergillus growth or reactive oxygen species (ROS) negatively impacts MSC survival. There are several licensing options whereby prior licensing of MSCs can be achieved, in vitro, prior to administration. Licensing options include proinflammatory cytokines, the anti-inflammatory cytokine TGF-β, via TLR ligand activation or by exposure of MSCs to disease microenvironments in the form of patient derived serum or bronchoalveolar lavage fluid (BALF). Image created using Biorender.com.

**Table 1 cells-10-02982-t001:** Pre-clinical models of lung disease that received licensed MSCs.

Murine Model	MSC Source	Route of Administration	Licensing Method	Licensing Location	Outcome	Ref
IPF (bleomycin)	Mouse BM-MSC	I.T.	Hypoxia (1.5% O_2_)	in vitro	↑ HIF1α, HGF, VEGF ↓ IL-6, pro-IL-1β	[67]
ALI	Rat BM-MSC	In perfusate	Hypoxia (1% O_2_)	in vitro	↑ IL-10, PGE2 ↓ Lung injury score ↓ TNF-α, IL-1β, MIP-2	[68]
ALI	Human UC-MSC	I.V.	TGF-β1	in vitro	↑ MSC survival ↑ Expression of RhoA ↓ LPS-induced injury	[69]
ALI	Human UC-MSC (EVs)	I.V.	IFN-γ	in vitro	↑ Animal survival ↑ eNOS ↓ Lung injury score ↓ TNF-α	[70]
ALI	Rat lung MSCs	I.V.	Culture on lung ECM-cyclic stretch	in vitro	↑ Lung elastance ↓ TNF-α, CXCL2 + neutrophils	[71]
ALI	Human UC-MSC	I.N.	Heatshock (42 °C for 1 h)	in vitro	↑ HSP70 expression ↑ IL-10 + PGE2 ↓ NLRP3 inflammasome formation ↓ IL-1β secretion by macrophages	[72]
ALI	Human BM-MSC	I.V.	Co-culture w/ serum from ARDS patients	ex vivo	↑ IL-10 + IL-1RN ↓ IL-6, IL-1α, IL-8, IL-1β, IFN-γ, TGFβ3, TGFβ2 + TNFAIP6	[73]
ALI	Human BM-MSC	AM treated with MSC- EVs-I.V.	Co-culture w/ BALF from ARDS patients	ex vivo	↑ M2 macrophage marker expression ↑ Phagocytic capacity of human MDMs ↓ Cytokine production	[74]
ALI	Human BM-MSC	I.T.	Cco-culture w/ plasma from ARDS patients	ex vivo	↑ IL-6 production ↑ N-cadherin expression at mRNA + protein levels ↓ CD105 + CD90 marker expression at day 5	[47]
ALI	Mouse BM-MSC	I.V.	Co-culture w/ serum from ALI mice	ex vivo	↑ Expression of anti-inflammatory mediators (TGF-β + IL-10) in AM in vitro ↓ iNOS + IL-6	[75]
Asthma	Mouse BM-MSC	I.T.	Co-culture w/ serum or BALF from asthmatic mice	ex vivo	↑ TGF-β, IFN-γ, IL-10,TSG-6, IDO-1, IL-1RN, iNOS, TNF-α, IL-1β + arginase-2 ↑ caspase-3, bax ↓ bcl-2Serum:↓ IL-4, IL-13 + eotaxin	[2]

Footnote ↑ = Increased, ↓ = Decreased. Idiopathic Pulmonary Fibrosis (IPF), Acute Lung Injury (ALI), Acute Respiratory Distress Syndrome (ARDS), Bone-Marrow Mesenchymal Stromal Cells (BM-MSC), Umbilical-Cord Mesenchymal Stromal Cells (UC-MSC), Intratracheal (I.T.), Intranasal (I.N.), Intravenous (I.V.), Alveolar Macrophages (AM), Extracellular Vesicles (EV), Extracellular Matrix (ECM), Bronchoalveolar Lavage Fluid (BALF).

**Table 2 cells-10-02982-t002:** In vivo studies utilising MSC licensing.

Murine Model	MSC Source	Route of Administration	Licensing Method	Licensing Location	Outcome	Ref
GvHD	Human BM-MSC	I.V.	IFN-γ	in vitro	↑ Animal survival ↓ Cellular infiltration ↓ Pathology severity in small intestine + liver	[43]
GvHD	Human MAPC	I.V.	IFN-γ / PPARδ antagonist	in vitro	↑ Efficacy on day 0 ↑ Retention in spleen + liver	[49]
GvHD	Human BM-MSC	I.V.	Cyclosporine A + IFN-γ	in vitro	↑ IDO production + activity ↑ Animal survival	[54]
GvHD	Human BM-MSC	I.V.	N/A	N/A	↑ Immunosuppression ↑ IDO production in recipient phagocytes	[76]
Corneal allograft	Mouse BM-MSC	I.V.	TGF-β1	in vitro	↑ CD73 expression ↑ Treg expansion ↑ Immunosuppression ↑ Allograft survival ↓ Syngeneic T cell proliferation	[77]
Corneal allograft	Rat BM-MSC	I.V.	IFN-γ + TNF-α + IL-1β	in vitro	↑ Nitric oxide production ↑ Suppression of syngeneic lymphocytes ↑ Allograft survival ↑ Myeloid cells in lung ↑ FoxP3^+^ Treg population in lung + spleen	[78]
Sepsis	BM-MSC	I.V.	N/A	N/A	↑ Animal survival ↑ Organ function ↓ TNF-α + IL-6	[79]
Arthritis	Human BM-MSC	I.V.	PPARβ/δ antagonist orIFN-γ + TNF-α	in vitro	↑ Immunosuppression ↑ NF-kB activity ↑ p65 binding on iNOS promoter ↓ Severity of disease	[48]
Myocardial Infarction	Human BM-MSC	I.V.	TNF-α	in vitro	↑ TSG-6 mRNA in lung ↓ Infarct size ↓ Pro-inflammatory proteases ↓ MMP9	[64]

Footnote: ↑ = Increased, ↓ = Decreased. Graft versus Host Disease (GvHD), Bone-Marrow Mesenchymal Stromal Cells (BM-MSC), Multipotent Adult Progenitor Cells (MAPC), Intravenous (I.V.).

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
