# Peer review of "The Inflammatory Lung Microenvironment; a Key Mediator in MSC Licensing"

_cells, 2021, doi:10.3390/cells10112982_

Round 1

Reviewer 1 Report

This is a well-written review of MSC treatment in inflammatory lung diseases. The topic is timely because of COVID-driven interest in using MSCs as a treatment for ARDS. The review focuses on the lung microenvironment in potentiating or lessening MSC therapeutic efficacy, which is increasingly recognised as an important factor in our understanding of MSCs. 

Some questions for consideration:

Licensing of MSCs with TLR ligands (p7) - What is thought to trigger from TLR2 vs TLR4 signalling that results in enhanced therapeutic efficacy in different disease settings? 

Differential effects of ARDS BALF on MSCs  in Section 5 (p8) -  The cell source for readouts in these studies were either MSCs (ref 120) or monocyte-derived macrophages (ref 86). Would this difference explain the seemingly contradictory findings (line 276 vs 280)? Perhaps different lung microenvironments affect host macrophages differently and this in turn impacts on how macrophages are influenced by MSC treatment.

Minor points:

Line 70 - "While earlier studies, did not..." (remove comma)

Lines 175-178 - "leads to increased    Production of IDO..." (remove paragraph)

Lines 273-277 - This is a long sentence that would be easier to understand if broken up.

Author Response

We thank the reviewer for their time and effort in providing a fair review of our manuscript. Below we have responded to the reviewers comments in a point- by-point response.

This is a well-written review of MSC treatment in inflammatory lung diseases. The topic is timely because of COVID-driven interest in using MSCs as a treatment for ARDS. The review focuses on the lung microenvironment in potentiating or lessening MSC therapeutic efficacy, which is increasingly recognised as an important factor in our understanding of MSCs. 

Some questions for consideration:

Licensing of MSCs with TLR ligands (p7) - What is thought to trigger from TLR2 vs TLR4 signalling that results in enhanced therapeutic efficacy in different disease settings? 

We thank the reviewer for this insightful question. We have added a little more discussion of studies comparing MSC licensing with TLR2/3/4. We have also added a sentence to address the question asked on page 7.

Differential effects of ARDS BALF on MSCs  in Section 5 (p8) -  The cell source for readouts in these studies were either MSCs (ref 120) or monocyte-derived macrophages (ref 86). Would this difference explain the seemingly contradictory findings (line 276 vs 280)? Perhaps different lung microenvironments affect host macrophages differently and this in turn impacts on how macrophages are influenced by MSC treatment.

We thank the reviewer for questioning this point. Indeed, we had made an error in citing reference 120. We have corrected this now and have included the correct reference.

Minor points:

Line 70 - "While earlier studies, did not..." (remove comma)

Lines 175-178 - "leads to increased    Production of IDO..." (remove paragraph)

Lines 273-277 - This is a long sentence that would be easier to understand if broken up.

We thank the reviewer for highlighting these errors. They have now been corrected/edited accordingly.

Reviewer 2 Report

This is a very interesting review dealing with how the microenvironment can affect MSC actions, and thus how MSC therapeutic activities can be altered (enhanced or hindered) by different disease conditions. The authors underline the importance of investigating the disease microenvironment at the time of MSC delivery in order to, for example, possibly understand the controversial effects exerted by MSCs administered in different phases or different severities in the same disease.

Below I report some observations that should be addressed by the authors in order for the manuscript to be considered for publication in Cells:

-   Abstract. “Recent clinical trials of mesenchymal stromal cell (MSC) therapy for paediatric acute graft  versus host disease (aGvHD) have highlighted the significant benefit to patients who respond to  MSC administration”. This sentence is too general and is not relevant to the topic of the review, I therefore suggest that the authors revise it.

-   In the Introduction section, the authors cited studies using [ref 4 and ref 5] (see line 37) bone marrow-derived MSCs in a bleomycin-induced animal model of pulmonary fibrosis, however other types of MSCs have been used in this model such as amniotic MSCs [Cargnoni et al, Stem Cells Transl Med. 2020 doi: 10.1002/sctm.20-0068], epithelial amniotic MSCs [Zhang & Lai, Stem Cell Research & Therapy 2020 doi.org/10.1186/s13287-020-01951-w] and umbilical cord MSC [Moroncini et al, PLoS ONE 2018, doi.org/10.1371/journal.pone.0196048]. Therefore, the introduction should not be limited to bone marrow-derived MSC but also include studies where MSC from other sources, such as perinatal tissues, have been employed in the context of bleomycin-induced animal model of pulmonary fibrosis.

-  To facilitate the reader in identifying the clinical trials described in the review, the authors should report the Clinical Trial Identifier in the text.

-   The authors should report/discuss the results from studies that have been published in peer reviewed scientific journals, and avoid citing those reported via press release (see line 104 and after)

-   Paragraphs 4, 5 and 6 (4. Licensing of MSCs enhances their therapeutic efficacy; 5. Exogenous licensing of MSCs; 6. Endogenous licensing of MSCs) do not have a clear structure and present overlapping information, making the manuscript difficult to read. In my opinion, to ease the reading , the paragraphs should be two instead of three, and should cover the following: 1) how the lung microenvironment may enhance MSC therapeutic efficacy and 2) how the lung microenvironment may negatively influence MSC licensing and therapeutic effects. Furthermore, in these paragraphs, the authors should underline that both in vitro and in vivo studies have been undertaken to understand the interactions between MSC and their microenvironment. In particular, the studies performed in vitro employed either exogenous factors (such as cytokines, chemokines, hypoxia, etc) or factors contained in biological samples from patients (such as bronchoalveolar lavage, serum, etc). On the other hand, the studies performed in vivo used either non-primed MSC to assess the impact of the endogenous microenvironment, or MSC primed with exogenous factors to modulate their properties (for example immunomodulatory or anti-apoptotic properties) in order to boost therapeutic efficacy of the cells.    

-   Line 206: “Licensing of UC-MSCs with oncostatin M enhanced MSC therapeutic efficacy in a bleomycin-induced fibrosis model” the authors should add the related reference.

-  Line 383 and followings: “Combined with data from chronic lung patients where MSCs have been investigated within clinical trials but have failed to demonstrate efficacy, and the evidence suggesting that MSCs cannot promote the regeneration of fibrotic tissue, it seems sensible that MSCs may be most suitable in acute inflammatory conditions”. The authors should better argue/discuss this claim reporting also some specific references.

Author Response

We thank the reviewer for their time and effort in providing a fair review of our manuscript. Below we have responded to the reviewers comments in a point by point response.

This is a very interesting review dealing with how the microenvironment can affect MSC actions, and thus how MSC therapeutic activities can be altered (enhanced or hindered) by different disease conditions. The authors underline the importance of investigating the disease microenvironment at the time of MSC delivery in order to, for example, possibly understand the controversial effects exerted by MSCs administered in different phases or different severities in the same disease.

Below I report some observations that should be addressed by the authors in order for the manuscript to be considered for publication in Cells:

-   Abstract. “Recent clinical trials of mesenchymal stromal cell (MSC) therapy for paediatric acute graft  versus host disease (aGvHD) have highlighted the significant benefit to patients who respond to  MSC administration”. This sentence is too general and is not relevant to the topic of the review, I therefore suggest that the authors revise it.

We are in agreement with the reviewer and have now edited the sentence to remove GvHD.

-   In the Introduction section, the authors cited studies using [ref 4 and ref 5] (see line 37) bone marrow-derived MSCs in a bleomycin-induced animal model of pulmonary fibrosis, however other types of MSCs have been used in this model such as amniotic MSCs [Cargnoni et al, Stem Cells Transl Med. 2020 doi: 10.1002/sctm.20-0068], epithelial amniotic MSCs [Zhang & Lai, Stem Cell Research & Therapy 2020 doi.org/10.1186/s13287-020-01951-w] and umbilical cord MSC [Moroncini et al, PLoS ONE 2018, doi.org/10.1371/journal.pone.0196048]. Therefore, the introduction should not be limited to bone marrow-derived MSC but also include studies where MSC from other sources, such as perinatal tissues, have been employed in the context of bleomycin-induced animal model of pulmonary fibrosis.

We thank the reviewer for highlighting this. We have now added more references to include additional MSC sources.

-  To facilitate the reader in identifying the clinical trials described in the review, the authors should report the Clinical Trial Identifier in the text.

We have now added the clinical trial identifiers throughout the text.

-   The authors should report/discuss the results from studies that have been published in peer reviewed scientific journals, and avoid citing those reported via press release (see line 104 and after)

We have removed the sentence and reference citing the press release.

-   Paragraphs 4, 5 and 6 (4. Licensing of MSCs enhances their therapeutic efficacy; 5. Exogenous licensing of MSCs; 6. Endogenous licensing of MSCs) do not have a clear structure and present overlapping information, making the manuscript difficult to read. In my opinion, to ease the reading , the paragraphs should be two instead of three, and should cover the following: 1) how the lung microenvironment may enhance MSC therapeutic efficacy and 2) how the lung microenvironment may negatively influence MSC licensing and therapeutic effects. Furthermore, in these paragraphs, the authors should underline that both in vitro and in vivo studies have been undertaken to understand the interactions between MSC and their microenvironment. In particular, the studies performed in vitro employed either exogenous factors (such as cytokines, chemokines, hypoxia, etc) or factors contained in biological samples from patients (such as bronchoalveolar lavage, serum, etc). On the other hand, the studies performed in vivo used either non-primed MSC to assess the impact of the endogenous microenvironment, or MSC primed with exogenous factors to modulate their properties (for example immunomodulatory or anti-apoptotic properties) in order to boost therapeutic efficacy of the cells.    

We have now moved some of the sentences from paragraph 4 which were better suited for discussion in paragraph 6 to enhance the reading. Indeed there is an element of overlap as the findings from a number of studies will provide important information supporting the arguments made in the three different sections. We have also added additional information on in vitro studies and in vivo studies comparing alternative licensing approaches. We thank the reviewer for their suggestions and feel that the changes made have enhanced the readability of these sections.

-   Line 206: “Licensing of UC-MSCs with oncostatin M enhanced MSC therapeutic efficacy in a bleomycin-induced fibrosis model” the authors should add the related reference.

We thank the reviewer for spotting this error. We note that this sentences refers to BM-MSCs and we have now inserted the appropriate reference.

-  Line 383 and followings: “Combined with data from chronic lung patients where MSCs have been investigated within clinical trials but have failed to demonstrate efficacy, and the evidence suggesting that MSCs cannot promote the regeneration of fibrotic tissue, it seems sensible that MSCs may be most suitable in acute inflammatory conditions”. The authors should better argue/discuss this claim reporting also some specific references.

We have now discussed this argument in more detail and added specific references.